# New Deletions in the *Hermansky-Pudlak Syndrome Type 5* Gene in a Japanese Patient

**Shinya Kato [1], Tsugumi Aoe [2], Akie Hamamoto [2], Hiroshi Takemori [3,\*] and Toshiya Nishikubo [4]**

[1] Department of Life Science and Chemistry, Graduate School of Natural Science and Technology, Gifu University, Yanagido 1-1, Gifu 501-1193, Japan; x4521031@edu.gifu-u.ac.jp

[2] Department of Chemistry and Biomolecular Science, Faculty of Engineering, Gifu University, Yanagido 1-1, Gifu 501-1193, Japan; gifu_u3032001@yahoo.co.jp (T.A.); ahama@gifu-u.ac.jp (A.H.)

[3] United Graduate School of Drug Discovery and Medical Information Sciences, Gifu University, Yanagido 1-1, Gifu 501-1193, Japan

[4] Division of Neonatal Intensive Care, Center of Maternal Fetal Medicine, Nara Medical University, 840 Shijo, Kashihara, Nara 634-8522, Japan; tttnishi@naramed-u.ac.jp

\* Correspondence: htake@gifu-u.ac.jp; Tel.: +81-58-230-7634

**Abstract:** The Hermansky-Pudlak syndrome (HPS) is a rare disease characterized by oculocutaneous albinism and prolonged bleeding. HPS is caused by alterations in *HPS1-10* and their related genes, comprising the biogenesis of lysosome-related organelles complex 1–3 and adapter protein 3. Here, we report a Japanese patient with HPS associated with mild hypopigmentation, nystagmus, and impaired visual acuity. Sequencing analyses of the mRNA of this patient revealed new deletions (ΔGA and ΔG) in the *HPS5* gene. This was the first case of *HPS5* gene deficiency in Japan, and the two above-mentioned deletions have not yet been reported among patients with HPS5.

**Keywords:** Hermansky-Pudlak syndrome type 5; oculocutaneous albinism; platelet aggregation; variant

## 1. Introduction

The Hermansky-Pudlak syndrome (HPS) was first reported in 1959, when two patients with oculocutaneous albinism, prolonged bleeding, and pigmented macrophages in the bone marrow were identified and their symptoms were described [1]. To date, genetic alterations (variants) in ten genes (*HPS1-10*) and related genes (*CHS1/LYST*) have been shown to be responsible for similar symptoms/phenotypes [2–5]. As the *HPS* gene products play an important role in the maturation of lysosome-related organelles (melanosomes, lytic granule, and primary granule) [6,7], patients with HPS share symptoms classified as lysosome-related diseases—Albinism, platelet dysfunction, and pulmonary fibrosis.

Melanosomes emerge from the trans-Golgi network and mature by the incorporation of constitutive proteins, such as the melanosome skeletal protein PMEL17 and the melanogenic enzyme tyrosinase [7,8]. These proteins are sorted into melanosomes, by four distinct trafficking protein complexes—Biogenesis of lysosome-related organelles complex 1–3 (BLOC-1–3) and adapter protein 3 (AP3) [7]. BLOC-1 is a multimeric complex, composed of HPS7, HPS8, and HPS9 [9]; BLOC-2 is composed of HPS3, HPS5, and HPS6 [10]; and BLOC-3 is composed of HPS1 and HPS4 [11]. The AP3 complex is composed of the products of four different gene products—AP3M1, AP3S1, AP3B1 (HPS2) and AP3D1 (HPS10) [5]. These complexes work in vehicle transport, as cargo proteins to endosomes and other target organelles [7].

A group of patients with HPS classified according to BLOCs, often share a similar severity of phenotype, which is also the same in model animals [6,12]. Patients having BLOC-3 deficiency,

in addition to hypopigmentation, exhibit severe phenotypes—Pulmonary fibrosis, hemorrhage, and granulomatous colitis. Patients with AP3 (HPS2) deficiency suffer from prolonged bleeding, recurrent respiratory infections associated with neutropenia, and nerve abnormalities (poor balance and conductive hearing loss). Patients with BLOC-1 deficiency show easy bruisability, hypopigmentation, and eye- and skin-related problems. Patients with BLOC-2 deficiency show a relatively mild phenotype, mainly hypopigmentation, and in some instances, colitis in HPS3 patients [13]. These patients with responsible variants have been reported worldwide; patients with HPS1, 4, 6, and 9 deficiencies have also been found in Japan [14].

## 2. Case Presentation Section

A 33-year-old Japanese female with oculocutaneous albinism, was examined at the Nara Medical University. She had mild hypopigmentation (see Figure 1A, hypopigmented hair), nystagmus, and impaired visual acuity. Blood biochemical values (Table 1) indicated that severe iron deficiency anemia was associated with a low number of erythrocytes and a low level of hemoglobin. Bleeding time (BT) was prolonged in some instances, when measured (4.5 min; occasionally 5–15 min in re-examinations). She was supplemented with 10 units of platelets, during parturition.

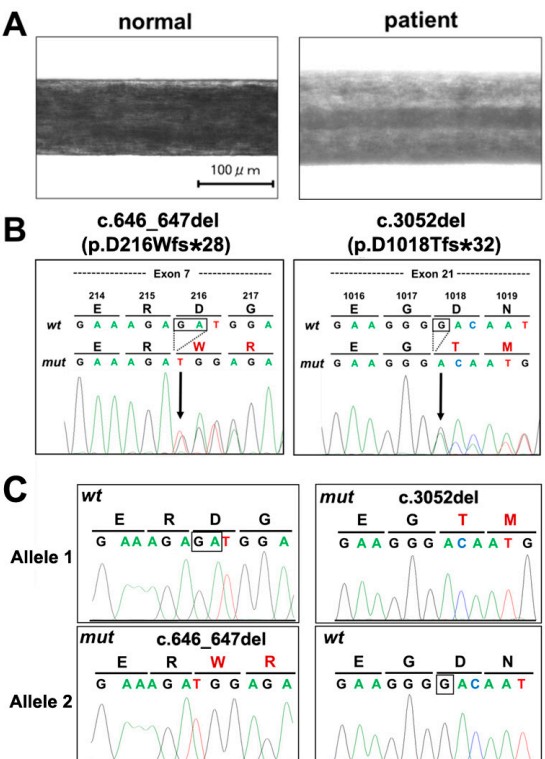

**Figure 1.** (**A**) Hair sample of a normal Japanese person and a patient with HPS5. (**B**) Sequence analyses of a Japanese patient with HPS-5. The 5′ variant is c.646_647del; p.D216Wfs*28 (**left**), and the 3′ variant is c.3052del; p.D1018Tfs*32 (**right**). Total RNA was isolated from leukocytes of a patient using the PureLink^TM RNA kit (Life Technologies, Carlsbad, CA, USA) and was used to synthesize cDNA using ReverTra Ace (TOYOBO, Osaka, Japan). After the amplification of the *HPS-5* gene fragments by PCR, variants were detected by direct sequencing (**B**) or after sub-cloning into plasmid vectors (**C**). Designs of primers are listed in Supplementary material. These deletions were also confirmed by genomic PCR, followed by sub-cloning into plasmid vectors (not shown). The protocols in this study were approved by the Nara Medical University, Nara, Japan (G130), and the Gifu University (29—349).

Platelet agglutination/aggregation tests were consistent with a Delta (δ)-storage pool disease, due to defects in intracellular organelles in platelets. Delta granules primarily contained calcium, ATP,

ADP, serotonin, histamine, and epinephrine. Kaolin did not induce ADP secretion from patient platelets. Platelet aggregation after treatment with a high concentration of collagen (5 μg/mL) was accompanied with prolonged lag times in the patient, and 1 μg/mL collagen failed to induce aggregation of patient platelets. This was also the case when arachidonic acid was used as a stimulant; 1 μM arachidonic acid partially induced platelet aggregation, which turned to dissociation, in a minute. Treatment with 0.2 μM arachidonic acid failed to induce aggregation of patient platelets. Thus, the patient was diagnosed with HPS.

**Table 1.** Hermansky-Pudlak syndrome (HPS) type 5 patient data obtained at 33 years of age.

| Hematology | | | Coagulation | | |
|---|---|---|---|---|---|
| WBC | $42 \times 10^2/\mu L$ | | sedimentation | 19s | |
| RBC | $409 \times 10^4/\mu L$ | | PT | 12.7s | |
| Neutrophil | 47.9% | | PT-INR | 1.12 | |
| Lymphocyte | 34.8% | | AP | 79% | |
| Monocyte | 12.7% | | APTT | 28.6 s | |
| Eosinophil | 2.7% | | fibrinogen | 271 mg/dL | |
| Basophil | 1.9% | | FDP | 2.4 μg/dL | |
| HGB | 6.8 g/dL | L | D dimer | 0.8 μg/dL | |
| HCT | 24.4% | L | BT | 4.5 m | |
| MCV | 59.7 fL | L | | | |
| MCH | 16.6% | L | | | |
| MCHC | 27.9% | L | | | |
| platelet | $21.7 \times 10^4/\mu L$ | | | | |
| **Biochemistry** | | | | | |
| albumin | 4.2 g/dL | | Na | 138 mEq/L | |
| globulin | 3.2 g/dL | | K | 4.1 mEq/L | |
| amylase | 30 U/L | | Cl | 103 mEq/L | |
| AST | 12 U/L | L | Ca | 9.0 mEq/L | |
| ALT | 7 U/L | L | Fe | 10 μg/dL | L |
| LDH | 127 U/L | | ferritin | <11.0 ng/ml | |
| ALP | 117 U/L | | TIBC | 393 μg/dL | L |
| γ-GTP | 10 U/L | | UIBC | 383 μg/dL | |
| TP | 7.4 g/dL | | CRP | 0.2 mg/dL | H |
| ALB | 4.2 g/dL | | eGFR | 101 mL/min | |
| Ch-E | 152 U/L | L | hemolysis | 0 | |
| blood glucose | 87 mg/dL | | milky fluid | 0 | |
| BUN | 10 mg/dL | | choloplania | 3 | |
| CRE | 0.57 mg/dL | | | | |
| CK | 23 U/L | L | | | |
| UA | 4.1 mg/dL | | | | |
| T-CHO | 153 mg/dL | | | | |
| TG | 110 mg/dL | | | | |
| T-Bil | 0.5 mg/dL | | | | |
| DB | 0.1 mg/dL | | | | |
| IB | 0.4 mg/dL | | | | |

WBC—white blood cell, RBC—red blood cell, HGB—hemoglobin, HCT—hematocrit, MCV—mean corpuscular volume, MCH—mean corpuscular hemoglobin, MCHC—mean corpuscular hemoglobin concentration, PT—prothrombin time, PT-INR—prothrombin time international normalized ratio, AP—active prothrombin, APTT—activated partial thromboplastin time, FDP—fibrin degradation products, AST—aspartate aminotransferase, ALT—alanine aminotransferase, LDH—lactate dehydrogenase, ALP—alkaline phosphatase, γ-GTP—gamma glutamyl transpeptidase, TP—total protein, ALB—albumin, Ch-E—cholinesterase, BUN—blood urea nitrogen, CRE—creatinine, CK—creatine kinase, UA—uric acid, T-CHO—total cholesterol, TG—triglycerides, T-Bil—total bilirubin, DB—direct bilirubin, IB—indirect bilirubin, Na—sodium, K—potassium, Cl—chloride, Ca—calcium, Fe—ferrum, TIBC—total iron binding capacity, UIBC—unsaturated iron binding capacity, CRP—C-reactive protein, eGFR—estimate glomerular filtration rate, L—low, H—high.

To identify the genetic alterations in the patient, we purified the total RNA from leukocytes, amplified cDNAs by reverse transcriptase-polymerase chain reaction, RT-PCR, and determined the sequences of coding regions for HPS-related genes, by direct sequencing. In consideration of the patient's pathological data, we suspected that she might have alterations in a gene for BLOC-2. When we examined the sequences of the 5′ regions of *HPS3*, *HPS5*, and *HPS6* genes, we found error peaks of nucleotides in the *HPS5* gene, suggesting the presence of deletions or insertions. Therefore, we carefully analyzed the *HPS5* gene.

As shown in Figure 1B, we identified novel deletions in the *HPS5* gene—A GA-deletion in the exon 7 and a G-deletion in the exon 21. The GA-deletion, del, at 646 and 647 nucleotides (coding [c.] DNA position 646_647; the sequence information was based on NM_181507), caused a frameshift (fs) with a conversion of the D–W amino acid (protein [p.]), following a premature termination codon (*) at 28th codon, c.646_647delGA (p.D216Wfs*28). The G-deletion also caused an fs, c.3052delG (p.D1018Tfs*32). Amplification of the exon 7 and exon 21 from genomic DNA, also supported that these deletions did not serve as artifacts during the synthesis of cDNA fragments (data not shown). By amplifying the whole coding region using a long PCR, followed by sub-cloning into a plasmid vector, we confirmed that these deletions were on the individual alleles (Figure 1C).

## 3. Discussion

Mild hypopigmentation, nystagmus, impaired visual acuity, and bleeding tendency associated with defects in platelet functions are common in patients with deficiencies of BLOC-2, including HPS5 [6,12]. In addition to these symptoms, the patient showed severe iron deficiency anemia.

Although iron deficiency is not a typical feature among HPS patients, some of them have been reported to share these symptoms with tendency of hemorrhages [15]. Iron has been found to promote melanogenesis in cultured retinal pigment cells. Iron supplementation upregulates the gene expression of melanogenic factors, such as the melanogenic transcription factor SOX10, and the enzymes tyrosinase and tyrosinase-related protein 1 [16]. Interestingly, the gene expression of the HPS3 protein, a component of BLOC-2, was also upregulated by iron, which was accompanied by an increase in the population of matured melanosome (Stage IV). These suggested that iron deficiency in the patient might augment the hypopigmented phenotype.

We previously reported that hypopigmented mice with *Hps5* gene deficiency, *ruby-eye 2* (*ru2*), tended to develop colitis [17], while patients with HPS1 and HPS4 also developed granulomatous colitis [16]. However, only a few patients with HPS5, as well as HPS3, have experienced similar gastrointestinal symptoms [2,12,13,18–22]. The Japanese patient in the present study also did not show such indications, suggesting that colitis associated with HPS5 deficiency might be a specific phenotype in mice.

Variant mapping in the human *HPS5* gene (Table 2) indicated that variants have been found throughout the coding region in this gene. This suggests that the whole structure of HPS5 protein might be required for the proper functioning of BLOC-2.

**Table 2.** Variants have been identified in the human *HPS5* gene.

| Exon | Previously Reported Variants [1] | | | New Variants |
|---|---|---|---|---|
| 3 | c.219G>A | | | |
| 5 | c.285-10A>G | c.302_305del | c.434G>A | |
| 7 | c.719G>C | c.818_822del | del 1.4 kb | c.646_647del |
| 8 | c.879dup | c.888_889insA | | |
| 12 | c.1417C>T | c.1423del | | |
| 13 | c.1618C>T | c.1634+1G>A | | |
| 16 | c.1871T>G c.1892T>C | c.1900del c.2026_2029del | c.2219T>C | |
| 18 | c.2593C>T | c.2624del | | |
| 19 | c.2750_2751del | | | |
| 20 | c.2926_2929dup | | | |
| 21 | c.2979_2982del | c.3034A>G | c.3058+3A>G | c.3052del |
| 22 | c.3096_3098del | | | |

The sequence information is based on NM_181507. [1] [21,22]. "c"—coding region, ">" indicates conversion, "_" indicates conversion in the intron upstream of the indicated exon, "+" indicates conversion in the intron downstream of the indicated exon, del—deletion, ins—insertion, dup—duplication, kb—kilo base. No mutation has been reported in exons 1–2, 4, 6, 9–11, 14–15, 17, or 23.

We note here that we have not examined whether the present variants in the *HPS5* gene were inherited or derived from de novo mutations. Moreover, we did not determine sequences of all pigmentation-related genes, including individual *HPS* genes. To confirm that the *HPS5* variants were the sole responsible causes for the patient's phenotypes, we have to determine the whole genome sequence in the future.

## 4. Conclusions

Only patients with *HPS1, 4, 6,* and *9* gene deficiencies have been found in Japan [14]. Thus, the *HPS5* gene deficiencies reported here in this patient were found in the first HPS type 5 case in Japan. The two *HPS5* gene deletions we found, have not been reported before, among patients with HPS5.

**Supplementary Materials:** The following are available online at http://www.mdpi.com/2571-841X/2/2/15/s1, Figure S1: Primer sequences of *HPS3, 5, 6* genes.

**Author Contributions:** S.K. and T.A. performed sequence analyses, A.H. and H.T. analyzed the sequence data, and T.N. made a diagnosis.

**Funding:** A part of this research was supported by grants from the Japanese Foundation for Applied Enzymology and the Ogawa Science and Technology Foundation.

**Conflicts of Interest:** The authors declare no conflict of interest.

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
