# Peer review of "New Deletions in the *Hermansky-Pudlak Syndrome Type 5* Gene in a Japanese Patient"

_reports, doi:10.3390/reports2020015_

Round 1
Reviewer 1 Report
Actually, there have been no reports of a patient with HPS5 among Japanese population.
However, for the accurate gene diagnosis for genetic diseases with Mendelian form of inheritance, authors should use genomic DNA at first.
Author Response
Thank you for your kind review. We could improve our manuscript.
Reviewer 1
Actually, there have been no reports of a patient with HPS5 among Japanese population.
However, for the accurate gene diagnosis for genetic diseases with Mendelian form of inheritance, authors should use genomic DNA at first.
(Response)
Thank you for your important suggestion. We had asked the patient’s parents to provide their samples for DNA sequencing. However, they rejected the request.
We performed genomic DNA sequencing and amplification of whole cDNA (ORF) of HPS5 mRNA and sub-cloning into a plasmid vector to examine that the two deletions were on different alleles.
We are sorry for the unclear description of these procedures. We rewrote the description in Line 105-108 (Results), and Line 137-141 (the end of Discussion), and added new figures in Fig. 1C.
Reviewer 2 Report
Originality:
The case report is interesting because it decribes the first case in Japan of a patient with two deletions (not described yet) in HPS5 gene.
Quality of Presentation:
There is a lot of spelling mistake and imperfections.
Line 31 There is an error on the gene name : related genes (CHS1/LIST LYST)
Line 56 She had mild hypopigmentation (Fig. 1A) >There is no picture of the patient.
Line 93 the author should use the HGVS nomenclature and the ACMG criteria to describe variants.
Line 98 How can the authors confirm that the pathogenic variants are in trans?
The authors should use the term variant instead of mutation.
The authors should better explain the diagnosis strategy (only with a RNA strategy? NGS Panel? Exome?)
Line 91 spelling mistake "genetic alternations that of the patient"
Line 119 the table must be completed with the HGVS nomenclature and with the reference publications.
Line 124 spelling mistake "cording region in this gene"
The article needs to be corrected and improved.
Author Response
Thank you for your kind review. We could improve our manuscript.
The case report is interesting because it decribes the first case in Japan of a patient with two deletions (not described yet) in HPS5 gene.
Quality of Presentation:
There is a lot of spelling mistake and imperfections.
(Response)
Thank you very much for your indications. We corrected the spelling.
Line 31 There is an error on the gene name : related genes (CHS1/LIST LYST)
Line 56 She had mild hypopigmentation (Fig. 1A) >There is no picture of the patient.
(Response)
The patient refused to be taken a photo. Therefore, we showed hair pigmentation. The original description was unclear. We rewrote the description (Line 57).
Line 93 the author should use the HGVS nomenclature and the ACMG criteria to describe variants.
(Response)
Thank you for your suggestion. We corrected “c.646_647del” and “c.3052del” “p.D216Wfs*28” and “p.D1018Tfs*32” Line 103-105, Fig. 1A and C, and Table 2.
Line 98 How can the authors confirm that the pathogenic variants are in trans?
(Response)
Thank you for your suggestion. We conformed by a long-PCR of the entire cording region followed by sub-cloning into a plasmid vector. We rewrote the description and added data in Fig. 1C. (Line 105-108)
The authors should use the term variant instead of mutation.
(Response)
Thank you for your suggestion. We used “variant” or “alteration” instead of “mutation”.
The authors should better explain the diagnosis strategy (only with a RNA strategy? NGS Panel? Exome?)
(Response)
We performed genomic DNA sequencing by exon amplification and whole amplification of the cording region of mRNA. We rewrote the descriptions. (Line 105-108).
Line 91 spelling mistake "genetic alternations that of the patient"
(Response)
Thank you for your indication. We corrected it. (Line 93).
Line 119 the table must be completed with the HGVS nomenclature and with the reference publications.
(Response)
Thank you for your suggestion. We rewrote the Table 2. And added reference No. (Line 133)
Line 124 spelling mistake "cording region in this gene"
The article needs to be corrected and improved.
(Response)
Thank you for your suggestion. (Line 130)
Reviewer 3 Report
Review of “New Deletions in The Hermansky-Pudlak Syndrome 3 Type 5 Gene in A Japanese Patient”
Introduction section
Line 36 - “Melanosomes emerge from the trans-Golgi network and maturate by the incorporation…” Change to “mature”, “maturate”is not a verb.
Line 38 – “These proteins are transferred into the melanosomes by a continuous movement of
four distinct trafficking protein complexes: biogenesis of lysosome-related organelles complex 1–3 (BLOC-1–3) and adapter protein 3 (AP3) [7].”
Should be changed to:
“These proteins are sorted into melanosomes by four distinct trafficking protein complexes: biogenesis of lysosome-related organelles complex 1–3 (BLOC-1–3) and adapter protein 3 (AP3) [7].”
Line 42 – “AP3 is a heterotetrameric complex composed of HPS2 and HPS10 proteins [5]. “
Should be changed to:
“The AP3 complex is composed of the products of four different genes: AP3M1, AP3S1, AP3B1 (HPS2) and AP3D1 (HPS10).”
Line 42-43 - “These complexes work as cargo proteins in vehicle transportation between endosomes and other target organelles [7].”
Should be changed to:
“These complexes work in vesicle transport of cargo proteins to endosomes and other target organelles.”
Comment on Line 50.
HPS3 patients have been reported to have colitis in childhood by Santiago Borrero et al (2006). Hence, colitis should be included in the manifestations of patients with BLOC2 deficiencies.
The following reference should be included in the introduction. Santiago Borrero PJ, Rodríguez-Pérez Y, Renta JY, Izquierdo NJ, Del Fierro L, Muñoz D, Molina NL, Ramírez S, Pagán-Mercado G, Ortíz I, Rivera-Caragol E, Spritz RA, Cadilla CL. Genetic testing for oculocutaneous albinism type 1 and 2 and Hermansky-Pudlak syndrome type 1 and 3 mutations in Puerto Rico. J Invest Dermatol. 2006, 126, 85-90.
Line 55 – “A 33-year-old Japanese female patient with the case of oculocutaneous albinism was examined at Nara Medical University.”
Should be re-written as “A 33-year-old Japanese female presenting with oculocutaneous albinism was examined at Nara Medical University”
Lines 58-59 “Prolonged bleeding was also anticipated until it stops…”
This sentence is not clear. Do the authors mean that prolonged bleeding times were expected? From the information provided in line 59 it appear that bleeding time was prolonged in some instances when measured.
Line 60 - “She had an experience of supplementation with 10 U platelet during parturition.”
Should be “She was supplemented with 10 units of platelets during parturition.” Since no abbreviations are provided, the “U” should not be abbreviated.
“Table 1. data at age 33 years.” Incomplete Table Title.
Suggest to change to “ HPS type 5 patient data obtained at 33 years of age.”
Table 1 has completely duplicated data (2 copies of same data), which needs to be removed
Abbreviations such as BT (bleeding time) should only be used after defining them.
Lines 73-75 “Platelet agglutination/aggregation tests suspected, the patient could be classified into δ-storage pool diseases that defected intracellular organelles in platelets: δ-granules containing primarily calcium, ATP, ADP, serotonin, histamine, and epinephrine”.
This sentence is too long, unclear and cumbersome:
Should be changed to: Platelet agglutination/aggregation tests were consistent with a Delta (δ)-storage pool disease due to defects in intracellular organelles in platelets. Delta granules contain primarily calcium, ATP, ADP, serotonin, histamine, and epinephrine.
Line 79 - “1 μM arachidonic acid partially induced platelet aggregation but which turned to dissociation in a”
The “but” is not needed.
Lines 80-81 – “0.2 μM arachidonic acid failed to induce aggregation of patient platelets. Thus, the patient was diagnosed as HPS.”
Corrected to sentence below, sentences should not start with a number:
"Treatment with 0.2 μM arachidonic acid failed to induce aggregation of patient platelets. Thus, the patient was diagnosed with HPS."
Line 91-93 – the words alteration and coding were misspelled (“alternation” and “cording”). In fact throughout the manuscript (Lines 93, 120 and 124), the word coding is misspelled consistently.
Correct to: "To identify genetic alterations in the patient we purified total RNA from leukocytes, amplified cDNAs by reverse transcriptase-polymerase chain reaction, RT-PCR, and sequences of coding regions for HPS-related genes determined by_________."
In addition, no information was provided in the methods section about how the RT-PCR products were sequenced. No mention is made if the mutations detected were also observed in the patient’s parents or were de novo mutations or in normal controls. No information was provided on any other detected genetic variations in the other HPS-related genes analyzed or which primers were used to carry out RT-PCR, at least the references used for the primer sequences should have been provided.
Line 101 – The words “Symptoms of” should be removed from this sentence, since mild hypopigmentation, nystagmus, impaired visual acuity, and bleeding tendency associated with defects in platelet functions are not symptoms, they are manifestation of HPS.
Line 114 – “We previously reported that hypopigmented mice with Hps5 gene deficiency, ruby-eye 2 (ru2) (Fig.1A), tended to develop colitis [14], while patients with HPS1 and HPS4 also developed granulomatous colitis [15].”
Several problems with this sentence: First, Figure 1A shows hair samples from a normal control and the HPS5 patient reported (not ru2 mice), and no mention of conclusions about these hair samples is made in the entire manuscript. Second, this sentence refers to ru2 mice, but reference 14 is a study on colitis in HPS patients, where they found colitis only in HPS1 and HPS4 patients. Reference 15 is a reference for the regulation by iron of the genes for tyrosinase, HPS3 and TRP1. No reference is given for the ru2 mice and colitis statement. I assume this is the correct reference, which needs to be added and reference numbers changed and referenced accordingly:
Itoh Y, Nagaoka Y, Katakura Y, Kawahara H, Takemori H. Simple chronic colitis model using hypopigmented mice with a Hermansky-Pudlak syndrome 5 gene mutation.
Pigment Cell Melanoma Res. 2016, 29,578-82.”
This sentence should be corrected in its entirety.
Line 117 - It is not correct to say that colitis has only been seen in mice with BLOC2 deficiencies. HPS3 patients have been reported to have colitis in childhood by Santiago Borrero et al (2006).
See previous comment in the Introduction section, for Line 50. This point should be addressed here.
Table 2 – Occupies almost 1 page, can be shortened by eliminating rows for exons 1-2, 4, 6, 9-11, 14-15, 17 and 23. A sentence can be added below the table indicating that no mutations have been reported in those exons.
Line 127-129 – “Thus, the HPS5 gene deficiency is the first case in Japan, and the above two deletions have also not been reported among patients with HPS5.
Correct to:
"Thus, the HPS5 gene deficiencies reported here in this patient were found in the first HPS type 5 case in Japan. The two HPS5 gene deletions we found have not been reported before among patients with HPS5."
Author Response
Thank you for your kind review. We could improve our manuscript.
Introduction section
Line 36 - “Melanosomes emerge from the trans-Golgi network and maturate by the incorporation…” Change to “mature”, “maturate”is not a verb.
(Response)
Thank you for your suggestion. We corrected the word. (Line 36)
Line 38 – “These proteins are transferred into the melanosomes by a continuous movement of four distinct trafficking protein complexes: biogenesis of lysosome-related organelles complex 1–3 (BLOC-1–3) and adapter protein 3 (AP3) [7].”
Should be changed to:
“These proteins are sorted into melanosomes by four distinct trafficking protein complexes: biogenesis of lysosome-related organelles complex 1–3 (BLOC-1–3) and adapter protein 3 (AP3) [7].”
(Response)
Thank you for your suggestion. We corrected the sentence. (Line 38-40)
Line 42 – “AP3 is a heterotetrameric complex composed of HPS2 and HPS10 proteins [5]. “
Should be changed to:
“The AP3 complex is composed of the products of four different genes: AP3M1, AP3S1, AP3B1 (HPS2) and AP3D1 (HPS10).”
(Response)
Thank you for your suggestion. We corrected the sentence. (Line 40-43)
Line 42-43 - “These complexes work as cargo proteins in vehicle transportation between endosomes and other target organelles [7].”
Should be changed to:
“These complexes work in vesicle transport of cargo proteins to endosomes and other target organelles.”
(Response)
Thank you for your suggestion. We corrected the sentence. (Line 43)
Comment on Line 50.
HPS3 patients have been reported to have colitis in childhood by Santiago Borrero et al (2006). Hence, colitis should be included in the manifestations of patients with BLOC2 deficiencies.
The following reference should be included in the introduction. Santiago Borrero PJ, Rodríguez-Pérez Y, Renta JY, Izquierdo NJ, Del Fierro L, Muñoz D, Molina NL, Ramírez S, Pagán-Mercado G, Ortíz I, Rivera-Caragol E, Spritz RA, Cadilla CL. Genetic testing for oculocutaneous albinism type 1 and 2 and Hermansky-Pudlak syndrome type 1 and 3 mutations in Puerto Rico. J Invest Dermatol. 2006, 126, 85-90.
(Response)
Thank you for your suggestion. We included information of HPS3 in Introduction and References. Line 52-53. Reference No. 13
Line 55 – “A 33-year-old Japanese female patient with the case of oculocutaneous albinism was examined at Nara Medical University.”
Should be re-written as “A 33-year-old Japanese female presenting with oculocutaneous albinism was examined at Nara Medical University”
(Response)
Thank you for your suggestion. We corrected the sentence. Line 56.
Lines 58-59 “Prolonged bleeding was also anticipated until it stops…”
This sentence is not clear. Do the authors mean that prolonged bleeding times were expected? From the information provided in line 59 it appear that bleeding time was prolonged in some instances when measured.
(Response)
Thank you for your suggestion. We corrected the sentence. Line 59 as “Bleeding time (BT) was prolonged in some instances when measured (4.5 min; occasionally 5 - 15 min in re-examinations).”.
(Line 59-61)
Line 60 - “She had an experience of supplementation with 10 U platelet during parturition.”
Should be “She was supplemented with 10 units of platelets during parturition.” Since no abbreviations are provided, the “U” should not be abbreviated.
(Response)
Thank you for your suggestion. We corrected the sentence. (Line 61)
“Table 1. data at age 33 years.” Incomplete Table Title.
Suggest to change to “ HPS type 5 patient data obtained at 33 years of age.”
(Response)
Thank you for your suggestion. We corrected the title. (Line 62)
Table 1 has completely duplicated data (2 copies of same data), which needs to be removed
Abbreviations such as BT (bleeding time) should only be used after defining them.
(Response)
Thank you for your suggestion. The duplication might be occurred in some word version(?). We re-checked by different PC including Mac-Win. We removed (BT).
Lines 73-75 “Platelet agglutination/aggregation tests suspected, the patient could be classified into δ-storage pool diseases that defected intracellular organelles in platelets: δ-granules containing primarily calcium, ATP, ADP, serotonin, histamine, and epinephrine”.
This sentence is too long, unclear and cumbersome:
Should be changed to: Platelet agglutination/aggregation tests were consistent with a Delta (δ)-storage pool disease due to defects in intracellular organelles in platelets. Delta granules contain primarily calcium, ATP, ADP, serotonin, histamine, and epinephrine.
(Response)
Thank you for your suggestion. We corrected the sentence. (Line 74-76)
Line 79 - “1 μM arachidonic acid partially induced platelet aggregation but which turned to dissociation in a”
The “but” is not needed.
(Response)
Thank you for your suggestion. We corrected the sentence. (Line 80)
Lines 80-81 – “0.2 μM arachidonic acid failed to induce aggregation of patient platelets. Thus, the patient was diagnosed as HPS.”
Corrected to sentence below, sentences should not start with a number:
"Treatment with 0.2 μM arachidonic acid failed to induce aggregation of patient platelets. Thus, the patient was diagnosed with HPS."
(Response)
Thank you for your suggestion. We corrected the sentence. (Line 81)
Line 91-93 – the words alteration and coding were misspelled (“alternation” and “cording”). In fact throughout the manuscript (Lines 93, 120 and 124), the word coding is misspelled consistently.
Correct to: "To identify genetic alterations in the patient we purified total RNA from leukocytes, amplified cDNAs by reverse transcriptase-polymerase chain reaction, RT-PCR, and sequences of coding regions for HPS-related genes determined by_________."
(Response)
Thank you for your suggestion. We corrected the sentence as “To identify genetic alterations in the patient we purified total RNA from leukocytes, amplified cDNAs by reverse transcriptase-polymerase chain reaction, RT-PCR, and determined sequences of coding regions for HPS-related genes by direct sequencing.” (Line 93-94)
In addition, no information was provided in the methods section about how the RT-PCR products were sequenced. No mention is made if the mutations detected were also observed in the patient’s parents or were de novo mutations or in normal controls. No information was provided on any other detected genetic variations in the other HPS-related genes analyzed or which primers were used to carry out RT-PCR, at least the references used for the primer sequences should have been provided.
(Response)
Thank you for your suggestion.
We are sorry for that we do not know whether the mutations were inherited mutations or de novo mutations, because the patient’s parent denied to provide samples for DNA-sequencing. We examined BLOC-2 due to her phenotypes. When 5’ regions were sequenced, we found a deletion in the HPS5 gene. Therefore, we focused on the HPS5 gene and found another deletion at the 3’ region. We prepared a primer list as supplemental material.
We added a sentence of “In consideration of the patient’s pathological data, we suspected that she might have alterations in a gene for BLOC-2. When we examined sequences of 5’ regions of HPS3, HPS5, and HPS6 genes, we found error peaks of nucleotides in the HPS5 gene, suggesting the presence of deletions or insertions. Therefore, we carefully analyzed the HPS5 gene. “ Line 105-108.
We added sentences at the end of Discussion. “We note here that we have not examined whether the present variants in the HPS5 gene are inherited or derived from de novo mutations. Moreover, we did not determine sequences of all pigmentation-related genes including individual HPS genes. To confirm the HPS5 variants are the sole responsible causes for the patient’s phenotypes, we have to determine whole genome sequence in the future.”
Line 137-141.
Line 101 – The words “Symptoms of” should be removed from this sentence, since mild hypopigmentation, nystagmus, impaired visual acuity, and bleeding tendency associated with defects in platelet functions are not symptoms, they are manifestation of HPS.
(Response)
Thank you for your suggestion. We removed the word. (Line 111)
Line 114 – “We previously reported that hypopigmented mice with Hps5 gene deficiency, ruby-eye 2 (ru2) (Fig.1A), tended to develop colitis [14], while patients with HPS1 and HPS4 also developed granulomatous colitis [15].”
Several problems with this sentence: First, Figure 1A shows hair samples from a normal control and the HPS5 patient reported (not ru2 mice), and no mention of conclusions about these hair samples is made in the entire manuscript. Second, this sentence refers to ru2 mice, but reference 14 is a study on colitis in HPS patients, where they found colitis only in HPS1 and HPS4 patients. Reference 15 is a reference for the regulation by iron of the genes for tyrosinase, HPS3 and TRP1. No reference is given for the ru2 mice and colitis statement. I assume this is the correct reference, which needs to be added and reference numbers changed and referenced accordingly:
Itoh Y, Nagaoka Y, Katakura Y, Kawahara H, Takemori H. Simple chronic colitis model using hypopigmented mice with a Hermansky-Pudlak syndrome 5 gene mutation.
Pigment Cell Melanoma Res. 2016, 29,578-82.”
(Response)
Thank you for your suggestion. We corrected the reference No.
This sentence should be corrected in its entirety.
Line 117 - It is not correct to say that colitis has only been seen in mice with BLOC2 deficiencies. HPS3 patients have been reported to have colitis in childhood by Santiago Borrero et al (2006).
See previous comment in the Introduction section, for Line 50. This point should be addressed here.
(Response)
Thank you for your suggestion. We corrected in Introduction and added References.
(Line 124)
Table 2 – Occupies almost 1 page, can be shortened by eliminating rows for exons 1-2, 4, 6, 9-11, 14-15, 17 and 23. A sentence can be added below the table indicating that no mutations have been reported in those exons.
(Response)
Thank you for your suggestion. We added a sentence as "No mutation has been reported in exons 1-2, 4, 6, 9-11, 14-15, 17 and 23. " (Line132-133)
Line 127-129 – “Thus, the HPS5 gene deficiency is the first case in Japan, and the above two deletions have also not been reported among patients with HPS5.
Correct to:
"Thus, the HPS5 gene deficiencies reported here in this patient were found in the first HPS type 5 case in Japan. The two HPS5 gene deletions we found have not been reported before among patients with HPS5."
(Response)
Thank you for your suggestion. We corrected the sentence. (Line 144-145)
Reviewer 4 Report
The authors report a novel compound heterozygous variant in HPS5 at first in a Japanese patient. Well written and explained.
Important changes:
In the abstract, HPS genes must be included because no all genes are HPS1-10… Moreover, this aspect must be change in the introduction section.
Authors must change the term mutations by variants.
Authors must include a paragraph about genetic analyisis of HPS, including that Next geneteration sequencing help us and improved the molecular diagnosis in this genetic heterogeneous disorder such as is explained in these 2 references (include) PMID: 30990103 and 28983057
Author Response
Thank you for your kind review. We could improve our manuscript.
The authors report a novel compound heterozygous variant in HPS5 at first in a Japanese patient. Well written and explained.
Important changes:
In the abstract, HPS genes must be included because no all genes are HPS1-10… Moreover, this aspect must be change in the introduction section.
(Response)
Thank you for your suggestion. We added words of “and their related genes” in Abstract. (Line 19).
Authors must change the term mutations by variants.
(Response)
Thank you for your suggestion. We changed to “varinats”
Authors must include a paragraph about genetic analyisis of HPS, including that Next geneteration sequencing help us and improved the molecular diagnosis in this genetic heterogeneous disorder such as is explained in these 2 references (include) PMID: 30990103 and 28983057
(Response)
Thank you for your important suggestion. Unfortunately, we do not have enough money to perform or order the analyses with Next generation sequencing. Therefore, we added sentences at the end of Discussion. (Line 137-141).
This report was just identification of new deletions in the HPS5 gene.